# EPA Modulates KLK Genes via miR-378: A Potential Therapy in Prostate Cancer

**DOI:** 10.3390/cancers14112813

**Published:** 2022-06-06

**Authors:** Kai-Jie Yu, De-Yi Ji, Ming-Li Hsieh, Cheng-Keng Chuang, See-Tong Pang, Wen-Hui Weng

**Affiliations:** 1Department of Chemical Engineering and Biotechnology, Graduate Institute of Biochemical and Biomedical Engineering, National Taipei University of Technology, Taipei City 106, Taiwan; m7398@cgmh.org.tw (K.-J.Y.); eversefmn@hotmail.com (D.-Y.J.); 2Department of Urology, Chang Gung Memorial Hospital, Tao-Yuan 333, Taiwan; h0810@cgmh.org.tw (M.-L.H.); ckchuang@gmail.com (C.-K.C.); 3College of Medicine, Chang Gung University, Tao-Yuan 333, Taiwan

**Keywords:** prostate cancer, EPA, KLK genes, miR-378

## Abstract

**Simple Summary:**

Castration resistant prostate cancer (CRPC) is one of the most pestilent form of prostate cancer (PCa), accounting for approximately 10–20% of all PCas, which carry poor mortality and morbidity rates. The focus in this study is on EPA, a natural supplement commonly found in fish oils. EPA induces the expression of PGC-1β gene and co-expresses miR-378 in cells, a nucleic acid sequence that inhibits PCa cell proliferation. This significance is highlighted in this study as a potential adjunctive therapy for all stages of PCa including CRPC.

**Abstract:**

It is known that miRNA-378a-3p (miR-378) could be induced by eicosapentaenoic acid (EPA), an omega-3 fatty acid. Herein, we first demonstrated how miR-378 exerts anti-prostate cancer (PCa) actions by influencing multiple target genes, including KLK2, KLK4, KLK6, and KLK14, which are implicated in PCa development, cell proliferation, and cell survival. Furthermore, these genes also correlate with androgen and mTOR signaling transduction, and are considered pivotal pathways for the onset and progression of PCa. In total, four PCa cell lines and eight pairing tissues (tumor vs. normal) from clinical PCa patients were included in the current study. The results showed high significance after EPA induced tumor cells containing higher expression levels of miR-378, and led the PCa cells having low cell viabilities, and they progressed to apoptosis when compared with normal prostate cells (*p* < 0.001). The findings indicated that EPA might become a potential therapy for PCa, especially because it is derived from the components of natural fish oil; it may prove to be a great help for solving the problem of castration-resistant prostate cancer (CRPC).

## 1. Introduction

Although omega-3 polyunsaturated fatty acids (omega-3 fatty acids), such as eicosapentaenoic acid (EPA) or docosahexaenoic acid (DHA), have been the subject of natural anticancer products, the effects and the possible underlying mechanisms of omega-3 fatty acid supplementation against cancer or cancer-related complications remain to be discussed [1]. We, therefore, focused on the investigation of correlations among microRNA-378a-3p (miR-378), which inhibits the progression of prostate cancer (PCa) through the suppression of kallikrein-related peptidase (KLK) genes, such as KLK2, KLK4, KLK6, and KLK14, and how EPA drives the expression level of miR-378 in cancer cells. Apparently, our current findings did reveal the partial mechanisms of how EPA protects PCa cells from cancer progression. Indeed, our previous colorectal cancer studies proved that the activities of molecular miR-378 could be successfully induced by EPA-stimulating gene PGC-1β transcription [2,3,4,5,6,7]. Herein, we hypothesized that miR-378 may play a crucial role in silencing the pivotal PCa-related genes, such as the KLK gene family, in which KLK3 is a gene encoding the glycoprotein enzyme in humans, which is a well-known prostate-specific antigen (PSA), and acts as the contemporary serum biomarker for PCa diagnosis. Nevertheless, the sequence of miR-378 was not capable of complementing the KLK3 gene sequence, which means that there are no chances to drive the expression of the KLK3 gene. However, the other KLK family genes, for example, KLK2, KLK4, KLK6, and KLK14, possess at least seven base pairs, which match with the molecular sequence of miR-378; this phenomenon provides considerable possibilities to suppress those genes’ activities, which would further affect the progression of PCa tumors (Figure 1).

KLKs are a protein family containing 15 genes, which are located at positions q13.33–q13.41 on chromosome 19 [8,9]. The high correlation between the androgen receptor (AR) and the KLK family has been proven in PCa [10,11,12,13]. Except for KLK3, the KLK2, KLK4, and KLK14 genes have been suggested to potentially correlate with AR signaling pathways, and they could be modulated by AR in the prostate [14,15,16]. Many reports have pinpointed that, although the expression of KLK2, KLK4, KLK6, and KLK14 might trigger Akt/MAPK signaling transduction, cell proliferation could be further stimulated through both protease-activated receptor 1 (PAR1) and protease-activated receptor 2 (PAR2) activities [11,17,18,19,20]. The overexpression of KLK2 and KLK4 genes is highly correlated with cancer occurrence, promoting cell invasion and/or cell metastasis [21,22,23,24,25,26]; the suggested mechanism indicates that KLK2 and KLK4 are involved in modulating insulin-like growth factor binding protein-3 (IGFBP-3) degradation in PCa, resulting in cell proliferation and cancer cell survival [11,27,28]. In addition, the KLK4 gene acts on the molecular circuit, involving the integration of androgen and mTOR signaling in PCa, mainly inhibiting the gene promyelocytic leukemia zinc finger (PLZF), inducing upregulation of the expression levels of AR and mTOR in PCa cells [15] (Figure 1). The gene KLK6 enhances the cell invasion ability through degradation of the ECM; therefore, it might also improve cell proliferation or cell migration [27,28]. Moreover, PCa is highly correlated with the MAPK/AKT pathway, from which KLK14 is intricately derived through AR signaling [16,25,26,29] (Figure 1).

The clinical parameters concerning miR-378 in PCa analysis have been reported by Margaritis Avgerish et al., who indicated that the KLK2 and KLK4 genes are highly correlated with miR-378 levels in PCa patients; particularly, a lower expression of miR-378 in PCa patients is significantly related to short-term recurrence when compared with control subjects (*p* value = 0.005) [30]. In addition, the low expression levels of miR-378 in PCa cells are significantly related to a high Gleason’s score, large tumor size, and increased serum PSA expression, increasing the overall risk of PCa tumor growth, with recurrence potentially being observed [30]. Further evidence, such as detailing the genes targeted by miR-378 in PCa, including PIK3CG, GRB2, AKT3, KLK4, and KLK14, has been presented by Sara Samaan et al. [9].

The novelty of the current study lies in the fact that we tried to modulate the genes KLK2, KLK4, KLK6, and KLK14 to inhibit the progression of PCa via miR-378, which can be produced in tumor cells through EPA, stimulating co-expression of the gene PGC-1β [2]. Hopefully, this will represent an efficient technique to inhibit castration-resistant prostate cancer (CRPC) cell growth, and further induce apoptosis in the cell.

## 2. Materials and Methods

### 2.1. Clinical Sample Collection

In total, eight pairs of tumor and normal FFPE tissues derived from PCa patients (including fourteen cases derived from prostatectomy, and two cases from needle biopsies) were obtained from Linkou Chang Gung Memorial Hospital, Taoyuan, Taiwan. All samples were collected under ethics committee approval: reference number 201800025B0C601. Five 5 µm thick consecutive tissue sections were prepared and reviewed by a pathologist to confirm the adequacy of the tumor cells and diagnosis. Tumor content was estimated by hematoxylin staining and used for further cancer relative IHC analysis.

### 2.2. PCa Cell Lines and Cell Culture

Five cell lines provided by Linkou Chang Gung Memorial Hospital were examined in this study, including four PCa cancer cell lines (22RV1, DU-145, LNCaP, and PC-3) and one immortalized normal prostate cell (PZ-HPV-7). The cell lines 22RV1, DU-145, and PC-3 contain a TP53 gene missense mutation; the LNCaP cell carries a gene AR missense mutation. All PCa cells were cultured in Roswell Park Memorial Institute 1640 Medium (Gibco, Grand Island, NY, USA) supplemented with 10% fetal bovine serum (Gibco, Grand Island, NY, USA). PZ-HPV-7 was cultured in serum-free medium Keratinocyte-SFM (Gibco, Grand Island, NY, USA) mixed with 20–30 μg/mL bovine pituitary extract (Gibco, Grand Island, NY, USA) and 0.1–0.2 ng/mL recombinant EGF (Gibco, Grand Island, NY, USA). Quantities of 1 mL/500 mL MycoZap^TM^ Plus-PR (LONZA, VZA-2021, Basel, Switzerland) antibiotics were added to each basal medium at 37 °C in a humidified atmosphere of 5% CO_2_. After harvesting, the cells were extracted for further mRNA, microRNA, protein, and cell viability analyses.

### 2.3. Elevating the Level of miR-378 in Cells through Incubation in EPA Medium

In order to indirectly stimulate cells to co-express miR-378 from its host gene PGC-1β by treatment with EPA (zikolakopoulou.2013), the four PCa cell lines and control cell line (PZ-HPV-7) were cultured in a 20 µM concentration of EPA mixed with culture medium for 24 h, in accordance with our previous studies [2]. After harvesting, the expression levels of miR-378 RNA were then measured by RT-PCT.

### 2.4. RNA Extraction and Quantification of miR-378

The expression level of miR-378 in cells was detected in original cells and was indirectly induced in EPA-treated cells, following the procedure detailed in our previous report, with minor modifications [2]. In brief, extraction of the total RNA was performed using the *mir*Vana™ miRNA Isolation Kit, measuring the cDNA of miR-378, following the TaqMan microRNA assay protocol (Applied Biosystems, Foster City, CA, USA). Quantitative real-time PCR (qRT-PCR) was performed using TaqMan Universal Master Mix II, with UNG (Applied Biosystems™, 4440038), and 20× miR-378 primer. The expression level of miR-378 was quantified by a comparative CT method with a Real-Time PCR System (StepOnePlus; Applied Biosystems), according to the manuals and protocols. The cDNA template (10 ng/15 μL) was amplified throughout the first stage, as follows: 1 min and 40 cycles of 15 s at 95 °C, then 1 min at 60 °C. siRNA-U6 was used as an endogenous control. Normalization of the total RNA extract was performed utilizing the PZ-HPV-7 cell line as a control in the experiment.

### 2.5. Detection of KLK Expression Levels before and after miR-378 Elevation

Total RNA in the PCa cell line was extracted using the *mir*Vana™ miRNA Isolation Kit (Invitrogen, Darmstadt, Germany). The forward and reverse primers of KLK2, KLK4, KLK6, and KLK14 genes were as follows: KLK2 forward 5′GGCTCTGGACAGGTGGTAAAGA3′, reverse 5′CGGTAATGCACCACCTTGGTGT3′; KLK4 forward 5′GGAACTCTTGCCTCGTTTCTGG3′, reverse 5′AGCGGGTCATAGAGCTTACTGC3′; KLK6 forward 5′GGTCCTTATCCATCCACTGTGG, reverse 5′GAACTCTCCCTTTGCCGAAGGT3′; and KLK14 forward 5′ GAGTGTCAGGCTGGGGAACTAT3′, reverse 5′AACTCCTGCACAGACCATGCCA3′. Quantitative real-time PCR (qRT-PCR) was carried out, using Smart Quant Green Master Mix (PROtech, Taipei, Taiwan). Subsequently, 10 ng of RNA was used to reverse the cDNA throughout the first stage, as follows: 10 min and 40 cycles of 15 s at 95 °C, then 1 min at 54 °C. The expression level was quantified as the relative quantitative (RQ) level, in which beta-actin was used as an endogenous control. Normalization of the total RNA extract was performed, utilizing the PZ-HPV-7 cell line as a control in the experiment. We further compared the mRNA expression of KLK genes before and after miR-378 was elevated.

### 2.6. Immunohistochemistry (IHC) Analysis

To further investigate the KLK proteins located on the PCa cells, pairs of clinical samples were collected and the distributions of proteins in the cytoplasm and nucleus were compared. Firstly, tumor cell locations were morphologically examined in all samples through hematoxylin staining. All FFPE sections were heated in an oven at 60 °C for 15 min. Then, they were washed three times for 10 min each time with xylene (ECHO, XA-2101, Taiwan), and rehydrated with 99%, 95%, and 75% ethanol and distilled water, and soaked in 1× TBST three times for 5 min; antigen retrieval was performed in 10 mM sodium citrate solution at 95 °C for 30 min, after cooling at room temperature and soaking in 1× TBST three times for 5 min. A hydrogen peroxide block kit was used to block the endogenous peroxidase activity for 5 min. After washing, protein blocking kits were used for 10 min at room temperature, soaking in 1× TBST three times for 5 min, and the monoclonal antibodies were applied to sections of antibodies with the following concentration dilutions of KLK2, KLK4, KLK6, and KLK14 (rabbit polyclonal): 1:300, 1:1000, 1:500, and 1:200, respectively, in 1× TBST buffer and with incubation for 16 h at 4 °C. Primary antibody enhance kits were applied at room temperature for 10 min. Diaminobenzidine tetrahydrochloride (DAB; ThermoFisher, Waltham, MA, USA) was used to demonstrate peroxidase activity. The slides were counterstained with Mayer’s hematoxylin, dehydrated, cleared, and mounted with DePex (BDH, Poole, Dorset, UK). Normal epithelium and glandular tissue served as internal positive controls when they were observed. In addition, strongly positive samples for each catenin from other series (kidney) were used as positive controls in each staining batch. After three blind observer reads, the H-scores were calculated according to the distribution of immune cells in all experimental groups. This was semi-quantitatively determined, as follows: 0 = negative; (+) = weakly positive; (++) = moderately positive; (+++) = strongly positive, and the formula of H-score = [1 × percentage of weakly staining cells + 2 × percentage of moderately staining cells + 3 × percentage of strongly staining cells].

### 2.7. Flow Cytometry Analysis

For the quantification of necroptotic, early, and late apoptotic cell populations, 0.5–1 × 106 cells were incubated in a 10 cm culture dish, with cell density reaching 70% when full. Cell lines were harvested with EPA medium; the control cell line was harvested by adding absolute ethanol, according to the protocol. After incubation for 24 h, cells were transferred into 1.5 mL reaction tubes using 1× TrypLE Express (Gibco, 12604-013). Cells were centrifuged at 1000 rpm for 5 min at 4 °C and the supernatant was aspirated; afterwards, cells were washed with 1 mL PBS twice and resuspended in 500 µL 1× binding buffer (FITC Annexin V Apoptosis Detection Kit I, BD Pharmingen™, 556547). Subsequently, 100 µL of the solution (1 × 10^5^ cells) was transferred to a 5 mL culture tube, and 5 μL of FITC Annexin V (51-65874X) and 5 μL propidium iodide (PI) staining solution (51-66211E) were added. Then, cells were gently vortexed and incubated for 15 min at RT (25 °C), protected from light. Next, 400 μL of 1× binding buffer was added to each tube, and the cells were analyzed using flow cytometry BD FACSCaliburTM (BD Biosciences, San Jose, CA, USA) within 1 h. A 488 nm argon ion laser was used for excitation; an FITC-Annexin V signal could be detected on the FL1 channel at the 518 nm wavelength, and a PI signal could be detected on the FL2 channel at the 620 nm wavelength. At least 10,000 single-cell events per sample were collected. FITC-Annexin V and PI showed the overlapping emission spectra. Color compensation was used to eliminate false-positive FITC fluorescence in the channel in which PI was acquired. Additional single-staining samples were prepared. FlowJo 7.6.2 was used for analysis. The gate setting was in accordance with FSC (forward scatter)/SSC (side scatter), charting the major cell area selected. Depending on the fluorescence intensity of FITC-Annexin V and PI, the populations could be distinguished into the following four quadrants: double-negative (healthy) cells, Annexin-V-positive (early apoptotic cells) cells, and double-positive (late apoptotic and necroptotic) cells. Using the image-based features intensity threshold (>30%) and contrast morphology for the PI channel, discrimination between late apoptotic (small fragmented nuclei) and necroptotic cells (non-fragmented enlarged nuclei) of the double-positive population was achieved.

### 2.8. Statistical Analysis

Each experiment was performed in triplicate. To eliminate background variation among the experiments, the PZ-HPV-7 cell line was used as a control group to normalize the two groups of cells treated with EPA medium and incubated with the control medium. Analyses and visualizations of flow cytometry data were performed with GraphPad Prism5 and FlowJo 7.6.2 software. The differences in miR-378 expression levels between PCa cells and normal prostate cells were analyzed by single-variant post hoc tests (ANOVE). A *p* value ≤ 0.05 was considered to indicate a statistically significant result.

## 3. Results

### 3.1. Comparison of miR-378 Expression between PCa Cells and Normal Prostate Cells before and after EPA Treatment

Four PCa cell lines (DU-145, LNCaP, PC-3, and 22RV1, established from CWR22R xenograft) and one immortalized normal prostate cell line (PZ-HPV-7, used as a control cell) were included in the current study. To compare the expression of miR-378 in the original normal cells and PCa cells, qRT-PCR was performed, which showed significantly lower levels in the PCa cells than the normal prostate cells (normalized to one in order to enable further comparison with other cells), with a *p* value < 0.05; moreover, after inducing miR-378 through treatment with EPA in all the cells for 24 h, the performance of miR-378 was significantly increased in all the cells, including the control cells (*p* value < 0.05), with the results shown in Figure 2.

### 3.2. Gene Expression of KLK2, KLK4, KLK6, and KLK14 in PCa Cells after EPA Treatment

To investigate the expression levels of the KLK2, KLK4, KLK6, and KLK14 genes in the cells, either treated with 20 µM EPA-containing medium in co-culture for 24 h or without treatment of EPA, the cells were measured with a quantitative real-time transcription polymerase chain reaction assay (qRT-PCR); the original control cell was normalized to one for further comparisons, and the results are shown in Figure 3. Clearly, the KLK2, KLK4, KLK6, and KLK14 gene expression levels are significantly higher in the original PCa cells compared to the control cells, with *p* value < 0.001. Interestingly, all the cells presented significantly decreasing expression levels after EPA treatment when compared to their original cells, with *p* value < 0.001 for all the cells; however, the control cell did not show significance in KLK4 and KLK14 gene expression after treatment with EPA in the cells. Our data show strong evidence that proves that increasing the expression level of miR-378 in cells by EPA induces the PGC-1β gene, which possibly affects KLK2, KLK4, KLK6, and KLK14 gene expression in cancer cells.

### 3.3. Immunostaining Analysis of KLK Proteins of Clinical Tissues

In total, eight pairs of formalin-fixed, paraffin-embedded (FFPE) tumor tissues and histologically normal tissues adjacent to the tumor were obtained from the same clinical PCa patients, and analyzed by immunostaining. All the tumoral samples confirmed the diagnosis of prostate adenocarcinoma through H&E staining, which was re-evaluated by a pathologist. Five areas were randomly selected, observed with a 20× objective lens, and blind read by three observers; the results were discussed when the findings were inconsistent (Figure 4).

All the KLK proteins were analyzed, and we compared the expressions of proteins between tumor cells and normal cells. In addition, we further evaluated the location of protein activities in the cytoplasm and nucleus of the cells. After statistical analysis, the results showed that the total protein expression in tumor cells was much higher than in normal cells, both in the cytoplasm and nucleus, with *p* value < 0.05. The median H-scores of KLK2, KLK4, KLK6, and KLK14 protein expression in the nuclei of normal cells were 9, 24, 12.5, and 14.5, respectively, as compared with the tumor cells (42, 73.5, 43.5, and 62, respectively); therefore, the *p* values were 0.0027, 0.0009, 0.0062, and 0.0009, respectively. Similarly, regarding the protein expressed in the cytoplasm, the median H-scores in normal tissues were 24, 30, 19.5, and 21, respectively, and in tumor tissues, they were 209, 197, 229.5, and 222, respectively; therefore, the *p* values were 0.0002, 0.0001, 0.0002, and 0.0002, respectively (Figure 5).

### 3.4. Flow Cytometric Evaluation of the Apoptosis and Necrosis of PCa Cells after Treated with EPA

The cells with or without treatment with EPA medium culture were harvested after 24 h and divided into the following two groups: EPA-treated and control, respectively. Flow cytometry analyses were then performed to observe all the cells (PZ-HPV-7, 22RV1, DU-145, LNCaP, and PC-3), and to analyze the distributions of cells, in order to distinguish whether the cell had died of apoptosis or necrosis. The results are presented in Figure 6, and, as expected, the control group of PZ-HPV-7 cells did not appear to undergo an apoptosis process, whereas, in all the PCa cells (22RV1, DU-145, LNCaP, and PC-3), early apoptosis was clearly observed (Figure 6A–D). This demonstrated that EPA could significantly induce PCa cell death by down-regulating the KLK2, KLK4, KLK6, and KLK14 genes in PCa cells, then triggering apoptosis in the cancer cells; however, a similar mechanism did not affect the normal cells.

### 3.5. Cell Viability Analysis of PCa Cells after Treatment with EPA

To evaluate cell proliferation, cell viability, and cytotoxicity, MTS assays were performed in all the PCa cells that were treated with 20 μM of EPA, and then compared with their original cells. Indeed, all the PCa cells tended to significantly decrease the cell viability, with *p* value < 0.05~0.001, except for the normal control cells (Figure 7).

## 4. Discussion

Acquired resistance to AR-targeted therapies has become the main contemporary problem in the treatment of clinical PCa patients. Our novel findings indicated that the current study used unsaturated fatty EPA to indirectly induce the expression of miR-378 in PCa cells, significantly resulting in the inhibition of KLK gene family (KLK2, KLK4, KLK6, and KLK14) activities in all the tested PCa cells, and further progressing the tumor cells towards apoptosis, without harming the normal tissues (*p* value < 0.05; Figure 7). These genes act as downstream factors in the AR signaling transduction pathway (Figure 1).

Most early stage PCa needs normal levels of testosterone to grow; however, castration-resistant prostate cancers (CRPCs) do not, but, instead, keep growing with very low levels of testosterone. However, usually after androgen deprivation, patients soon acquire resistance to AR-targeted therapies, and virtually all patients will eventually progress to CRPC. Most novel hormone therapies targeting AR either inhibit the synthesis of androgens or block AR activities, such as abiraterone and enzalutamide, respectively, and, thus, have significantly improved the survival of metastatic CRPC (mCRPC) patients [31,32]. However, there is a considerable proportion of patients who present with primary resistance to these agents, whereas the secondary mutation formed might trigger tumor progression into the metastasis stage, eventually resulting in a high rate of lethality [33]. Therefore, we aimed to determine another possible way to try and block the gene located downstream of AR; miR-378 could be a suitable molecular target to inhibit PCa cell progression.

Nutrition interventions support cancer patients, and these forms of treatment are commonly accepted in the primary or secondary prevention of many types of cancer [34] However, limited reports have provided evidence outlining the mechanism of how EPA inhibits cancer progression [35]. To the best of our knowledge, this is the first study to detail the EPA inhibition mechanisms of KLK2, KLK4, KLK6, and KLK14 gene activities, consequently prohibiting PCa cell proliferation. This could be because EPA efficiently induces PGC-1β gene activity, resulting in the co-expression of miR-378 in tumor cells; the correlated mechanisms have been outlined in our previous report [33]. The miR-378 sequence is located at the 5′ end of the PGC-1β mRNA sequence; therefore, this small molecule further silences its targeting genes, such as KLK2, KLK4, KLK6, KLK14, etc. All the results showed that this significantly induced cell apoptosis in the PCa cells, which was proven in cell viability, cell death, and apoptosis assay experiments (*p* < 0.05) (Figure 6). In addition, the IHC analysis demonstrated that all the KLK protein expressions were much higher in clinical tumor cell tissue plasma and nuclei than in normal tissues (*p* < 0.001) (Figure 5). All the evidence strongly suggests that miR-378 plays a very important role in inhibiting downstream genes of the AR signaling pathway, and could be a potential molecule to treat PCa-resistant cases.

To prove the effect of miR-378 in PCa cells, the following two methods were performed to increase the expression of miR-378 in cancer cells: (i) the direct transfection of miR-378a-3p; (ii) the indirect induction of miR-378a-3p through EPA-stimulating PGC1-beta gene transcription. The results demonstrated that lower cell viabilities were strongly linked to higher expression levels of miR-378 in PCa cells (*p* value < 0.05; Figure 6). Based on these results, we provide evidence that miR-378 does play a crucial role in inhibiting PCa tumor proliferation, and it might be increased through EPA stimulation in PCa tumor cells, which further downregulates the activities of KLK genes to block the AR signaling pathway. Although the PCa cell lines used in the current study exhibited different AR statuses, after increasing the level of miR-378 in the cells, they all significantly suppressed KLK genes, and might further indirectly modulate AR signaling transduction as a feedback inhibition effect (Figure 1 and Figure 3). In addition, the microRNA miR-378a-5p, which complements miR-378a-3p (the sequence used in this study), might be induced by EPA at same time; therefore, the possibility of signal conduction should always be considered [34]. Hopefully, through our findings, this small molecular miR-378 might represent a novel and effective adjuvant for clinical treatments, and provide a solution to the current dilemma of PCa, in order to achieve better clinical results.

## 5. Conclusions

Indeed, EPAs are considered immunonutrients and are commonly used in the nutritional therapy of cancer patients, due to their considerable biological effects [1]. Here, we demonstrate that EPA plays an essential role in inducing the activity of the PGC-1β gene, leading to miR-378 co-expression in PCa cells; therefore, KLK gene signaling transduction can be suppressed, inducing the progression of apoptosis in PCa cells. These findings might provide clinical and therapeutic guidance for treating CRPC patients.

## Figures and Tables

**Figure 1 cancers-14-02813-f001:**
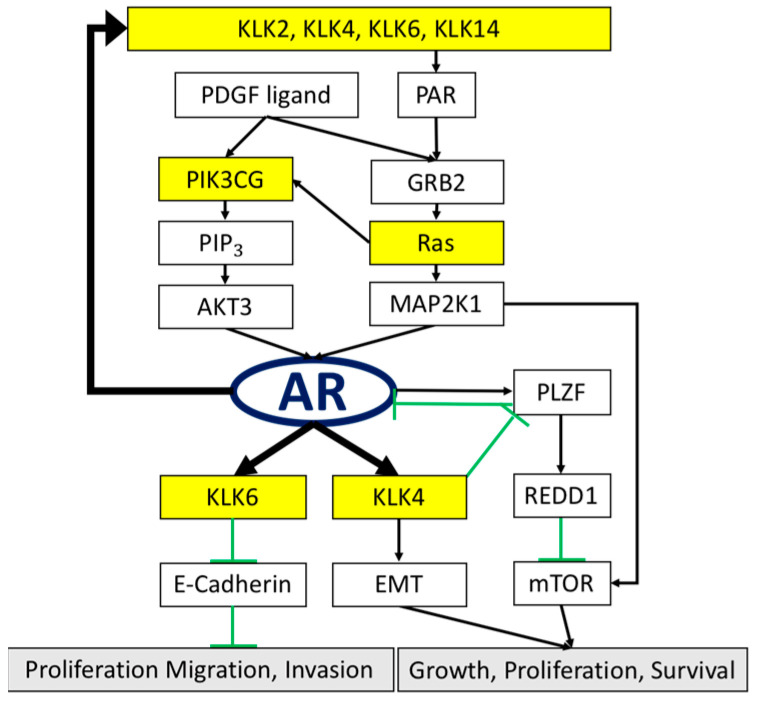
The schematic predicts the therapeutic efficacy of miR-378 silencing KLK genes in prostate cancer (PCa). The genes highlighted with a yellow background represent the gene sequences that partially complement miR-378 with at least 7 base pairs that sufficiently modulate their expression levels. The arrows represent promotion of the downstream gene activity, and the T sign (green) represents inhibition of their downstream genes. The AR gene might modulate the genes KLK2, KLK4, KLK6, and KLK14, and then feed back to itself through the PAR and the downstream RAS gene pathway.

**Figure 2 cancers-14-02813-f002:**
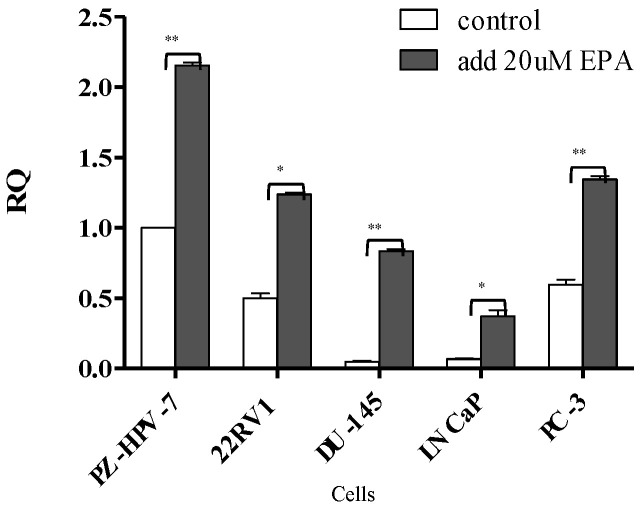
Comparison of the miR-378 expression levels before and after EPA treatment in each cell, measured by quantitative real-time polymerase chain reaction (qRT-PCR). EPA untreated (white bar) and EPA treated (black bar). *Y*-axis risk quotient (RQ) = tumor cells/reference sample. The PZ-HPV-7 cell was used as a reference sample, and its quality was normalized to 1, then further compared with other treated or untreated EPA cells. The data presented in the figure are the mean ± standard deviation. * represents *p* < 0.05, ** represents *p* < 0.005.

**Figure 3 cancers-14-02813-f003:**
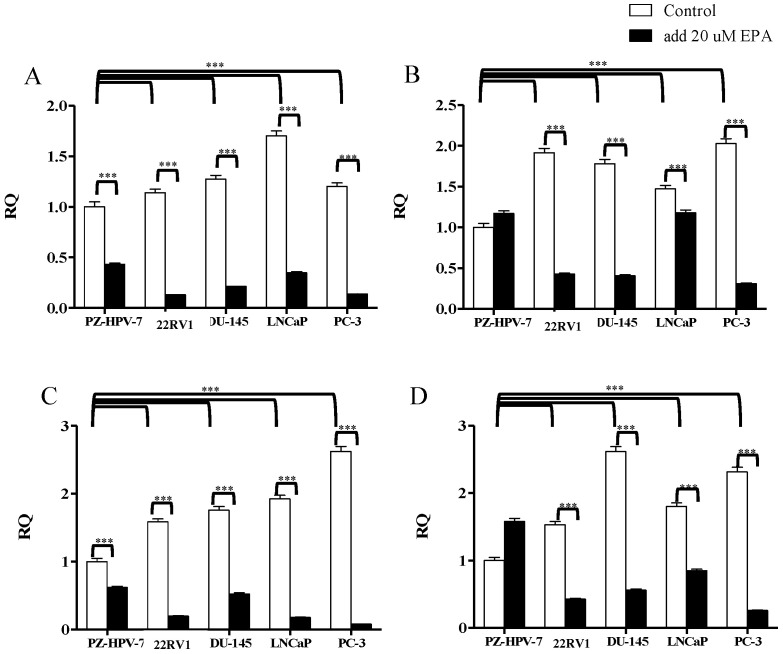
Comparison of the relative expression of KLK gene expression in each cell before and after treatment with EPA, measured by qRT-PCR. White bars represent original cells and black bars represent cells after treatment with 20 µM of EPA. *Y*-axis risk quotient (RQ) = tumor cells/reference sample. The PZ-HPV-7 cell was used as a reference sample, and its number quality was normalized to 1, then further compared with other cells. All the cells were also compared to their original cells. (**A**) KLK2 gene expression, (**B**) KLK4 gene expression, (**C**) KLK6 gene expression, and (**D**) KLK14 gene expression. The data presented in the figure are the mean ± standard deviation. *** represents *p* < 0.001.

**Figure 4 cancers-14-02813-f004:**
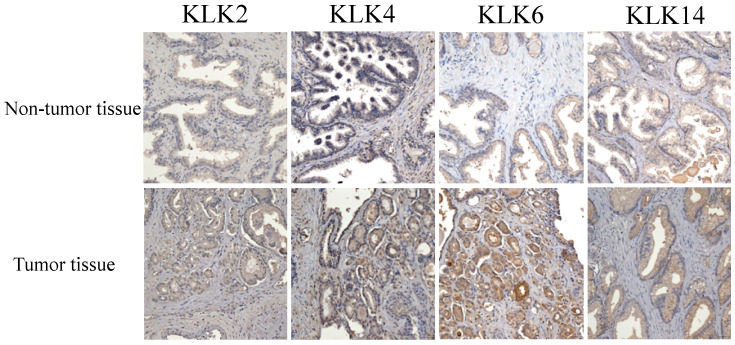
The immunostaining of KLK proteins showed significant differences in PCa tissues. The results show that the KLK2, KLK4, KLK6, and KLK14 proteins were significantly elevated compared with normal tissues in the nucleus and cytoplasm. The scale of the picture is 50 µm with a 2.0× objective lens.

**Figure 5 cancers-14-02813-f005:**
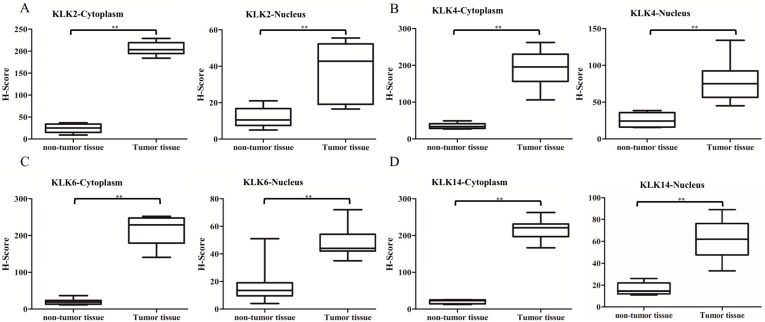
Significant KLK protein expression exhibited in PCa tissues either in the cytoplasm or nucleus. Box diagram of protein expression in surrounding normal tissues and tumor tissues. (**A**) KLK2; (**B**) KLK4; (**C**) KLK6; (**D**) KLK14. The data in the figure are the mean ± standard deviation. ** represents *p* < 0.001.

**Figure 6 cancers-14-02813-f006:**
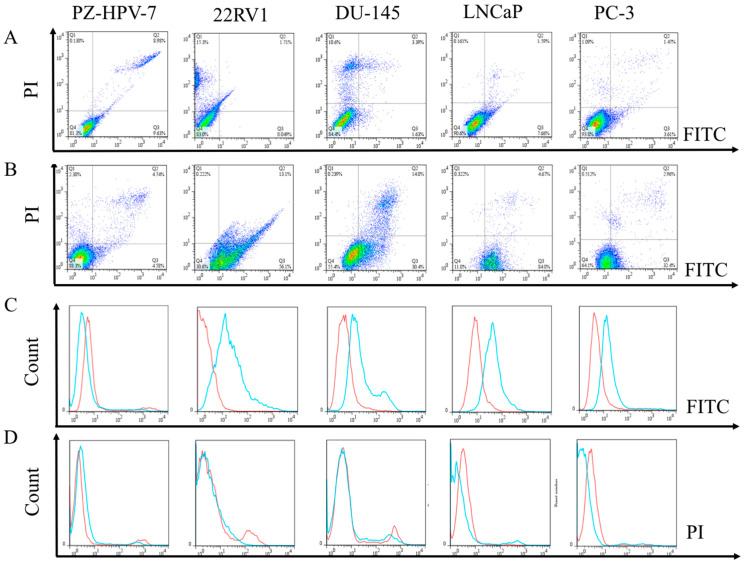
Flow cytometric analysis indicating that most of the tumor cells progressed towards apoptosis. (**A**) Original cells. (**B**) Cells after treatment with 20 µM of EPA. (**C**) The intensity of fluorescein isothiocyanate (FITC) labelled on cells shows the degree of cell apoptosis; red curves represent the original cells and blue curves represent cells after treatment with EPA—this is also shown on the right-hand side of the four-quadrant axes in (**A**,**B**). (**D**) The intensity of propidium iodide (PI) shows the degree of cell necrosis; red curves represent the original cells and blue curves represent cells after treatment with EPA—this is also shown on the upper side of the four-quadrant axes in A and B. Cells located in Q1 or Q2 zones tended towards necrosis; cells in the bottom right of the four-quadrant axis (Q3 zone) tended towards early apoptosis; cells in the Q4 zone were healthy.

**Figure 7 cancers-14-02813-f007:**
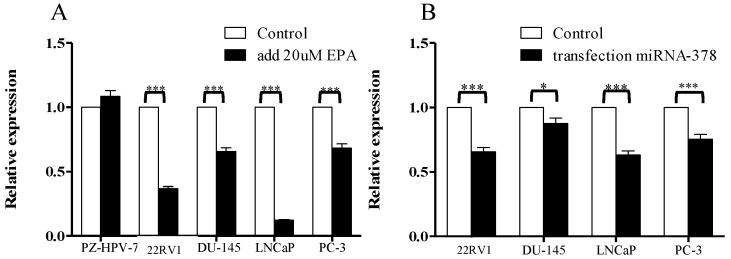
Cell viability significantly decreased after PCa cells were treated with EPA. (**A**) After the addition of 20 µM of EPA to all the PCa cells, the flow cytometry analysis indicated significantly increased cell death; (**B**) MTS analysis showed that after transfection with mimic miR-378, the cells also presented significant cell death. * represents *p* value < 0.05; *** represents *p* value < 0.001.

## Data Availability

The data presented in this study are available on request from the corresponding author.

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
