# Peer review of "EPA Modulates KLK Genes via miR-378: A Potential Therapy in Prostate Cancer"

_cancers, 2022, doi:10.3390/cancers14112813_

Round 1
Reviewer 1 Report
Reviewer Report
Yu et al. EPA modulates KLK genes via miR-378- A potential therapy in prostate cancer. Manuscript ID: cancers-1464235
In this study, the authors report one of the omega-3 fatty acids, eicosapentaenoic acid (EPA) as a potential therapy for the prostate cancer (PCa) treatment. The authors aimed to investigate underlying mechanisms of omega-3 fatty acids supplementation against cancer or cancer-related complications. They reported to demonstrate that miR-378 is induced by EPA and has anti-prostate cancer actions through silencing the kal-likrein-related peptidases (KLK)-2, -4, -6 and -14 genes that are implicated in the progression of PCa.
In general, the topic is interesting as it addressed the natural compound omega-3 polyunsaturated fatty acids, such as EPA treatment for prostate cancer.
However, there are some major concerns that are listed below:
Throughout the manuscript there are some language issues including 1) multiple typos, 2) confusing statements (e.g. Introduction: lines 91-100; lines 101-104) and 3) long sentences e.g. abstract lines 23-28 which include parts of the methods and results description. Thus extensive editing of English language and style required. Some statements incorrectly referenced and some statements are not referenced. Most Figure legends are missing titles. Some data requires statistical analyses. Study limitations are not discussed.
Major concerns:
- Introduction and Discussion contain unclear and sometimes confusing writing e.g. Introduction: lines 91-100 and lines 101-104: ‘specially the expression of miR-378 in PCa patients is significantly lower than normal people of the short-term recurrence when compare with the normal people (p value = 0.005).’ Thus, Introduction and Discussion should require major editing.
- Figure 1: It would be good to include a description of genes/pathways shown in schematic diagram. Last statement in the figure 1 legend: ‘From our results also proved the tumor cell viability significantly decrease with p< 0.001. .’ does not appear to indicate any direct relationship to the data shown in the figure.
- CWR22R is the name of xenograft; the cell line name is 22Rv1 (https://www.atcc.org/products/crl-2505)
- The authors selected PCa cell lines with different AR status. However, with the focus being on AR signalling this appears not to be discussed.
- Throughout the manuscript there is a confusion regarding what are ‘original normal cells’, ‘original control cells‘, ‘original PCa cells’, ‘normal control cells’ and ‘control cells’. These should be corrected and consistent throughout the manuscript e.g. normal PCa cells should refer to PZ-HPV-7 cells and untreated control cells refer to all cells that did not undergo treatment.
- It is not clear why the EPA treatment was at 20uM.
- Most Figure legends do not have titles. The title of a figure legend should describe the figure, clearly and succinctly. A title may summarize the result or major finding that you are drawing from the data in the figure.
- The figures should include a statistical test(s) performed and the number of technical repeats when applicable.
- Lines: 133-136 ‘Our data showed… miR-378… can effectively and directly inhibit KLK... genes…’ is overrepresented. Knockdown experiments of miR-378 should be performed to support this statement.
- For the Western blot results (Results 2.2), western blot gel images including loading controls should be included in the figure or as part of supplementary material to support the Western blot results.
- Flow cytometry data (Results 2.4) requires statistical analyses.
Minor concerns:
- Line 69. Reference 14 is incorrect and appropriate reference is required for ‘genes KLK2, KLK4 and KLK14 which have been also mentioned might correlated with AR signaling pathways, and they could be modulated by AR in prostate [14].’
- Line 61-62: ‘physiological procedures’ should this be physiological processes?
- Line 62 ‘… blood pressure adjustment, semen liquefaction ability etc.’ the ‘etc’ is not correct to state.
- line 116: showed in Figure 2 (Figure 2) these duplications should be deleted and throughout the manuscript.
- Description of abbreviations should take place when it is first mentioned e.g.. qRT-PCR lines 111 and 127.
- Figures 2 and 3 Y-axis RQ abbreviation should be described in the figure legend.
- Line 160 the origin of ‘normal tissues’ is not described? It should be clarified whether these are histologically normal tissue adjacent to the tumour obtained from the same patient.
Author Response
Attachment file

Reviewer 2 Report
The manuscript is well written and clear, the amount and quality of work are commendable, and the proposed approach can be useful for the scientific community.
Study protocol is designed properly with all aspects to rule out methodological biases.
Before final decision a minor changes should be incorporated into manuscript:
- introduction -is too detailed, especially lines 55-80. Should be shortened.
- some minor linguistic correction
Author Response
Attachment file

Round 2
Reviewer 1 Report
I am writing to report that my comments were satisfactorily addressed in the revised version of the manuscript by Yu et al. EPA modulates KLK genes via miR-378- A potential therapy in prostate cancer. Manuscript ID: cancers-1464235.
The legend is required for the Supplementary figure 1. The correction of typos is still required.